# A terpene synthase-cytochrome P450 cluster in *Dictyostelium discoideum* produces a novel trisnorsesquiterpene

Xinlu Chen[1], Katrin Luck[2], Patrick Rabe[3], Christopher QD Dinh[4], Gad Shaulsky[5], David R Nelson[6], Jonathan Gershenzon[2], Jeroen S Dickschat[3], Tobias G Köllner[2], Feng Chen[1]*

[1]Department of Plant Sciences, University of Tennessee, Knoxville, United States; [2]Department of Biochemistry, Max Planck Institute for Chemical Ecology, Jena, Germany; [3]Kekulé-Institute of Organic Chemistry and Biochemistry, University of Bonn, Bonn, Germany; [4]Department of Biochemistry and Molecular Biology, Baylor College of Medicine, Houston, United States; [5]Department of Molecular and Human Genetics, Baylor College of Medicine, Houston, United States; [6]Department of Microbiology, Immunology, and Biochemistry, University of Tennessee Health Science Center, Memphis, United States

**Abstract** Terpenoids are enormously diverse, but our knowledge of their biosynthesis and functions is limited. Here we report on a terpene synthase (*DdTPS8*)-cytochrome P450 (*CYP521A1*) gene cluster that produces a novel C12 trisnorsesquiterpene and affects the development of *Dictyostelium discoideum*. DdTPS8 catalyzes the formation of a sesquiterpene discoidol, which is undetectable from the volatile bouquet of wild type *D. discoideum*. Interestingly, a *DdTPS8* knockout mutant lacks not only discoidol, but also a putative trisnorsesquiterpene. This compound was hypothesized to be derived from discoidol via cytochrome P450 (CYP)-catalyzed oxidative cleavage. *CYP521A1*, which is clustered with *DdTPS8*, was identified as a top candidate. Biochemical assays demonstrated that CYP521A1 catalyzes the conversion of discoidol to a novel trisnorsesquiterpene named discodiene. The *DdTPS8* knockout mutant exhibited slow progression in development. This study points to the untapped diversity of natural products made by *D. discoideum*, which may have diverse roles in its development and chemical ecology.
DOI: https://doi.org/10.7554/eLife.44352.001

*For correspondence: fengc@utk.edu

Competing interests: The authors declare that no competing interests exist.

## Introduction

With over 80,000 structures identified, terpenoids constitute the largest class of natural products made by living organisms (*Christianson, 2017*). Most terpenoid natural products are known from plants (*Chen et al., 2011*), but bacteria (*Yamada et al., 2015*; *Dickschat, 2016*) and fungi (*Keller et al., 2005*; *Schmidt-Dannert, 2015*) are also rich sources. Recently we have shown that dictyostelid social amoebae, a class of eukaryotic soil microorganisms, also have the genetic capacity to produce monoterpenes ($C_{10}$), sesquiterpenes ($C_{15}$) and diterpenes ($C_{20}$) (*Chen et al., 2016*). Dictyostelid social amoebae have a unique life cycle, consisting of both unicellular and multicellular phases. When their bacterial food supply becomes scarce, amoebae start to aggregate, going through clearly-defined morphological changes to eventually form fruiting bodies (*Kessin, 2001*). *Dictyostelium discoideum* and *D. purpureum* are among the most extensively investigated species of dictyostelid social amoebae. Our recent studies illustrated that the multicellular stages of the life cycle of both *D. discoideum* (*Chen et al., 2016*) and *D. purpureum* (*Chen et al., 2018*) are characterized by the emission of a complex mixture of volatile organic compounds that is dominated by

sesquiterpenes. While some terpenoids are produced by both species, most appear to be species-specific (*Chen et al., 2018*). The biological function of these terpenoids is completely unknown.

The *D. discoideum* genome contains nine full-length genes (*DdTPS1-9*) encoding terpene synthases, the pivotal enzymes that create the terpenoid carbon skeleton (*Christianson, 2017*). All nine genes are expressed during the multicellular stage, suggesting a role for volatile terpenoid biosynthesis in fruiting body development. Indeed, biochemical characterization of the encoded proteins DdTPS1-9 revealed terpene synthase activity for all tested enzymes. The sesquiterpene products of DdTPS1-4, DdTPS6, DdTPS7 and DdTPS9, as well as the diterpene product of DdTPS5, could be detected in the volatile bouquet of *D. discoideum* during multicellular development (*Chen et al., 2016*; *Rabe et al., 2016a*; *Rinkel et al., 2017*). *D. discoideum* also released the monoterpene linalool that was produced by several recombinant terpene synthases including DdTPS2, DdTPS3 and DdTPS9, suggesting that at least one of these enzymes functions as a monoterpene synthase in vivo. In brief, the in vitro products of all DdTPSs except DdTPS8 could be detected in the volatile bouquet of *D. discoideum*.

DdTPS8 has sesquiterpene synthase activity, but does not produce mono- or diterpenes in vitro (*Chen et al., 2016*). *DdTPS8* exhibits the second highest level of expression among all nine *DdTPS* genes during development, so it was surprising to see that the major sesquiterpene product of *DdTPS8* was not detected in the *D. discoideum* volatiles (*Chen et al., 2016*). One possibility is that the product of DdTPS8 serves as a substrate for other enzymes, particularly cytochrome P450s (CYPs). CYPs are heme-containing proteins that catalyze a wide variety of oxidative reactions (*Coon, 2005*), often on terpenoid substrates (*Hamberger and Bak, 2013*). In this context it is interesting to note that the profile of volatiles emitted by *D. discoideum* contains a number of unknown compounds that are candidates for derivatives of the DdTPS8 product. Here we present conclusive evidence that the DdTPS8 sesquiterpene product is modified by a CYP that is physically clustered with DdTPS8 on chromosome six and produces a novel trisnorsesquiterpene named as discodiene. A *DdTPS8*-knockout mutant of *D. discoideum*, which failed to produce discodiene, showed slower progression in development than the wild type strain. This raises interesting questions about the specific biological role of discodiene as one constituent of a bouquet of volatiles produced by *D. discoideum* during its development.

## Results

### The product of DdTPS8 is the new sesquiterpene alcohol discoidol

DdTPS8 had been shown to have sesquiterpene synthase activity in previous work (*Chen et al., 2016*), but the structure of its product was not identified. In the present study, the coding sequence of DdTPS8 was cloned into the expression vector pET32a and expressed heterologously in *Escherichia coli*. After protein purification by $Ni^{2+}$-NTA affinity chromatography, the recombinant DdTPS8 enzyme was incubated with its substrate farnesyl diphosphate (FDP), resulting in the formation of a sesquiterpene alcohol as a single product. The compound was purified and structure elucidation by NMR spectroscopy (*Supplementary file 1*) revealed a new bicyclic sesquiterpenoid, which was named discoidol (*Figure 1A*). Discoidol is a stereoisomer of the known sesquiterpene alcohol jinkoheremol (*Nakanishi et al., 1983*).

The absolute configuration of discoidol was determined by enzymatic conversion of enantioselectively deuterated (*R*)- and (*S*)-(1-$^{13}$C,1-$^{2}$H)geranyl diphosphate (*Rabe et al., 2017*) that were elongated with isopentenyl diphosphate to the corresponding FDP isotopomers using the FDP synthase from *Streptomyces coelicolor* (*Rabe et al., 2016b*). Their conversion by the discoidol synthase resulted in enantioselectively deuterated discoidol with known absolute configuration at the deuterated carbon (*Figure 1B*). The absolute configuration of discoidol was then deduced by assignment of the relative orientation of the two hydrogen atoms at C2 by NOESY (*Figure 1C*). The additional $^{13}$C-NMR label at this carbon was used for a highly efficient analysis by HSQC spectroscopy, giving an intensive cross-peak for the attached hydrogen (*Figure 1—figure supplement 1*). These experiments resulted in the assigned absolute configuration of (4*S*,5*S*,7*S*)-discoidol.

The proposed cyclization mechanism of discoidol synthase (*Figure 1D*) starts with a 1,10-cyclization of FDP with attack of water to yield the neutral intermediate hedycaryol. A protonation-induced second cyclization results in cation 1, that upon a 1,2-hydride shift to cation 2, 1,2-methyl group

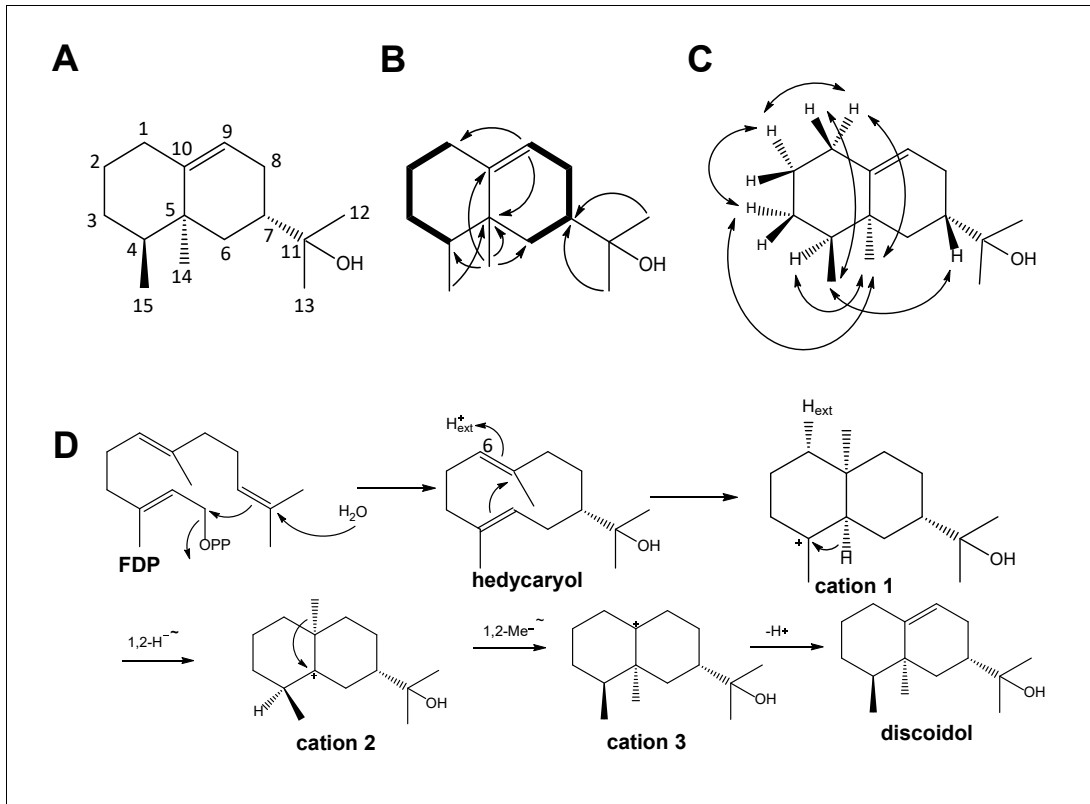

**Figure 1.** Structure elucidation of DdTPS8 product and biosynthetic mechanism. (**A**) Structure of discoidol. (**B**) Contiguous spin systems indicated by bold lines observed in discoidol by 1H,1H-COSY NMR, single headed arrows indicate diagnostic HMBC correlations. (**C**) important NOESY correlations that are indicated by double headed arrows observed in discoidol. (**D**) biosynthetic mechanism from farnesyl diphosphate (FDP) to discoidol catalyzed by DdTPS8. See also *Figure 1—figure supplements 1–4*.

DOI: https://doi.org/10.7554/eLife.44352.002

The following figure supplements are available for figure 1:

**Figure supplement 1.** Determination of the absolute configuration of discoidol.
DOI: https://doi.org/10.7554/eLife.44352.003
**Figure supplement 2.** Investigation of the reprotonation step in the cyclization mechanism of discoidol.
DOI: https://doi.org/10.7554/eLife.44352.004
**Figure supplement 3.** Investigation of the 1,2-hydride migration by incubation of (3–13C,2–2H)FDP with DdTPS8.
DOI: https://doi.org/10.7554/eLife.44352.005
**Figure supplement 4.** Incubation experiments with (12–13C)FDP and (13–13C)FDP and DdTPS8.
DOI: https://doi.org/10.7554/eLife.44352.006

migration to cation three and deprotonation reacts to form discoidol. This mechanism was experimentally supported by a series of incubation experiments using isotopically labeled substrates. The reprotonation of hedycaryol was demonstrated by incubation of (6-$^{13}$C)FDP (*Rabe et al., 2015*) with discoidol synthase in a deuterium oxide enriched buffer, resulting in a triplet in the $^{13}$C-NMR spectrum of the product cation 1 (*Figure 1—figure supplement 2*) that indicated a direct $^{13}$C-$^2$H connection. The stereochemical course for the reprotonation was evident from HSQC analysis of the obtained labeled product (*Rabe et al., 2016b*), demonstrating reprotonation from the *Si* face of C6. The 1,2-hydride shift from cation 1 to cation two was followed with (3-$^{13}$C,2-$^2$H)FDP (*Klapschinski et al., 2016*), resulting in a triplet for C4 of discoidol that demonstrated a direct $^{13}$C-$^2$H bond (*Figure 1—figure supplement 3*). Finally, the stereochemical fate of the terminal geminal methyl groups of FDP was followed with (12-$^{13}$C)FDP and (13-$^{13}$C)FDP (*Rabe et al., 2015*), showing a stereospecific incorporation of labeling into discoidol (*Figure 1—figure supplement 4*).

## A *DdTPS8* knockout mutant of *D. discoideum* lacks discoidol and a putative discoidol metabolite

The absence of the DdTPS8 product discoidol in *D. discoideum* culture coupled with the high level of expression of the corresponding gene, suggested that discoidol is produced but further modified in vivo. If this hypothesis holds true, the disruption of discoidol biosynthesis would abolish the biosynthesis of the modified product as well. Therefore, we analyzed a *D. discoideum* mutant in which *DdTPS8* was disrupted by an insertion of a blasticidin resistance cassette between nucleotides 544 and 545 of the *DdTPS8* open reading frame (*Figure 2A*), which was verified by sequencing (*Figure 2—figure supplement 1*). The *DdTPS8* mutant was next allowed to develop until the culmination stage of multicellular development (the ultimate stage in fruiting body formation when the expression of *DdTPS8* is the highest) and subjected to headspace chemical profiling. In comparison

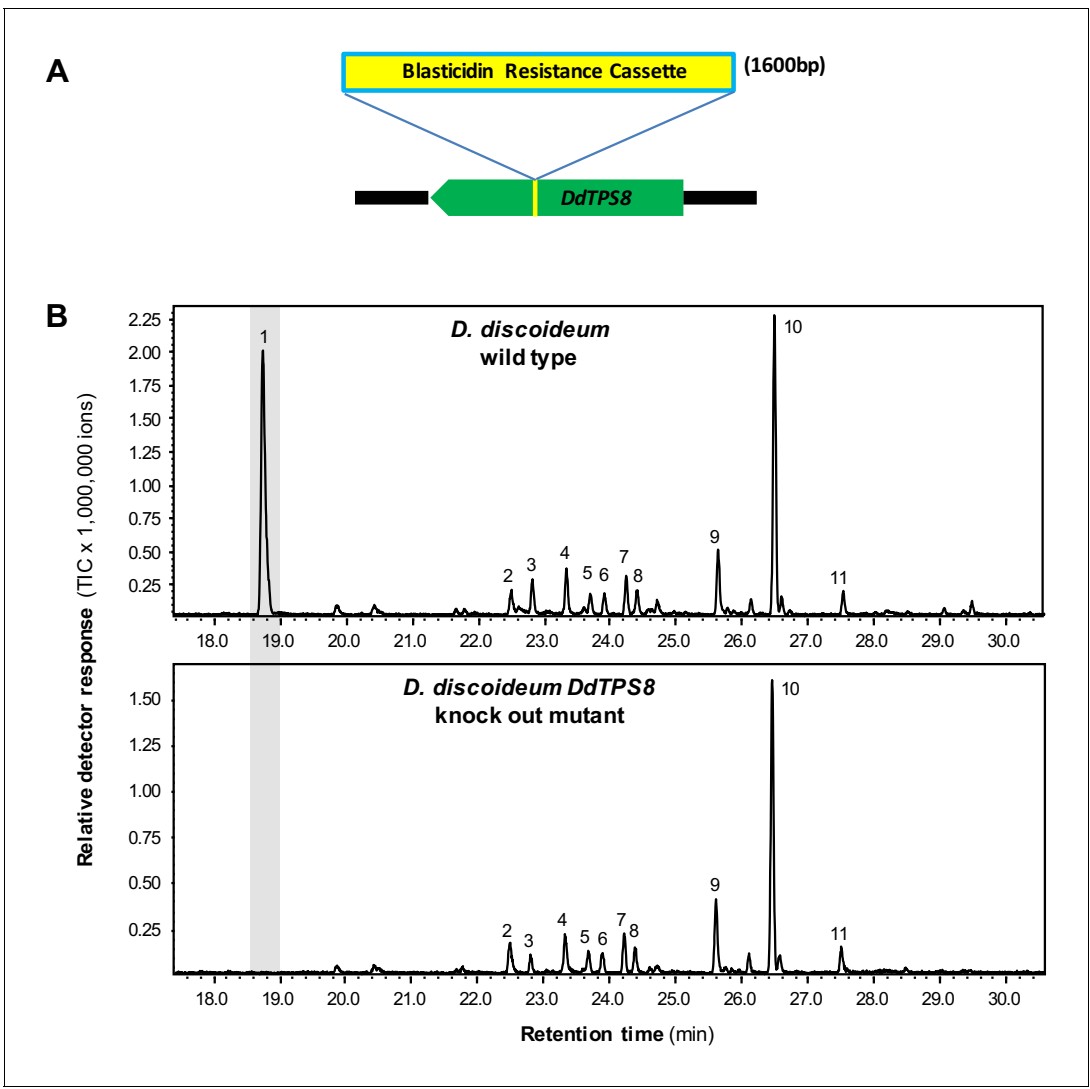

**Figure 2.** *DdTPS8* insertional mutant and its volatile profile. (A) Schematic presentation of *DdTPS8* gene with an insert of 1.6 kb. (B) Volatiles were collected from the headspace of the cultures and analyzed using GC-MS. Total ion chromatograms are shown. 1, unknown compound; 2, unidentified compound; 3, unidentified sesquiterpene hydrocarbon; 4, unidentified compound; 5, β-maaliene; 6, aristolene; 7, calarene; 8–10, unidentified sesquiterpene hydrocarbons; 11, nerolidol. See also *Figure 2—figure supplement 1*.

DOI: https://doi.org/10.7554/eLife.44352.007

The following figure supplement is available for figure 2:

**Figure supplement 1.** Verification of *DdTPS8* insertion mutant.

DOI: https://doi.org/10.7554/eLife.44352.008

to the wild type, the *DdTPS8* mutant lacked a major peak (peak one in *Figure 2B*) whose structure was unknown. This unknown compound appeared to have a molecular mass of 162, consistent with the mass of a degraded sesquiterpene that had lost a fragment containing three carbon atoms and one oxygen atom. Based on these findings, we hypothesized that the unknown compound is a putative C12 trisnorsesquiterpene derived from discoidol through C-C bond cleavage.

## The CYP family of *D. discoideum*: identification and coexpression analysis with *DdTPS8*

Among the diverse reactions catalyzed by CYPs with terpenes as substrates are oxidative degradations (*Stanjek et al., 1999*; *Irmler et al., 2000*; *Larbat et al., 2009*; *Lee et al., 2010*). This led us to hypothesize that the cleavage of discoidol to form the unknown compound is catalyzed by a CYP. Analysis of the *D. discoideum* genome led to the identification of a total of 54 putative *CYP* genes. Among them, 41 were annotated as full-length intact genes whereas the rest were pseudogenes or partial genes (*Supplementary file 2*). The 41 full-length genes were assigned to 17 families and 34 subfamilies (*Supplementary file 2*).

Coexpression analysis of *TPS* genes and *CYP* genes has been a useful tool to identify candidate CYPs that catalyze the modification of TPS products (e.g. *Ginglinger et al., 2013*). Thus we performed coexpression analysis of *CYP* genes with *DdTPS8* in *D. discoideum* based on their expression patterns during the 24 hr of multicellular development that consists of several stages: vegetative growth, streaming, loose aggregate, mound, Mexican hat, and fruiting body (*Figure 3A*). Gene expression data of *CYP* genes and *DdTPS8* were obtained from dictyExpress (http://dictyexpress.biolab.si) (*Parikh et al., 2010*) and used to calculate Pearson correlation coefficients (r) between individual *CYP* genes and *DdTPS8*. Among the 41 *CYP* genes, three showed significant correlation coefficients with *DdTPS8* exhibiting a P value lower than 0.001 (*Supplementary file 3*): *CYP521A1* (r = 0.994), *CYP508C1* (r = 0.992) and *CYP519C1* (r = 0.951). These three *CYP* genes, like *DdTPS8*, showed a maximal level of expression at 16 hr during multicellular development (*Figure 3A*). Examination of gene expression during development revealed that the transcripts of both *DdTPS8* and *CYP521A1* were nearly undetectable in vegetatively growing cells. Small amounts of transcripts accumulated between 4–8 hr of development, continued to accumulate until they peaked at 16 hr, and declined thereafter (*Figure 3A*). We also noticed that the top candidate *CYP521A1* is located 685 bp away from *DdTPS8* in a head-to-head configuration on chromosome 6 (*Figure 3B*), suggesting a possibility that *DdTPS8* and *CYP521A1* form a biosynthetic cluster.

## CYP521A1 catalyzes the oxidative degradation of discoidol to form the novel trisnorsesquiterpene discodiene

In our first attempts to characterize the enzymatic activity of CYP521A1, we expressed it together with yeast or *Arabidopsis* P450 reductases in *Saccharomyces cerevisiae* and incubated the resulting microsome preparations with discoidol. However, no enzymatic activity was detected. Assays containing yeast microsomes, recombinant DdTPS8 produced in *Escherichia coli*, and (*E,E*)-FDP showed no activity either. The lack of activity could be due to the incompatibility of the P450 reductase from a plant (*Arabidopsis*) or a fungus (yeast) with a CYP from *D. discoideum*. As such, we turned to a P450 reductase from *D. discoideum*. Among the three P450 reductase genes, *redA*, *redB* and *redC*, of *D. discoideum*, *redB* showed an expression pattern (*Gonzalez-Kristeller et al., 2008*) similar to that of *DdTPS8* and *CYP521A1*. Thus, we selected *redB* for our assays. The open reading frames of *CYP521A1* and *redB* were inserted into the vector pRSFDuet−1, which allows their coexpression in *E. coli*. The resulting constructs, together with another expression vector carrying the complete open reading frame of *DdTPS8*, were both introduced into *E. coli* Bl21 (DE3)-Star. Assuming that the intrinsic FDP pool of *E. coli* is sufficient to provide substrate for DdTPS8, we analyzed potential terpene accumulation in the headspace of the resulting *E. coli* culture (*Figure 4A*). Indeed, GC-MS analysis confirmed the formation of both discoidol with a molecular mass of *m/z* = 222 (*Figure 4B*) and the unknown terpenoid with a molecular mass of *m/z* = 162 (*Figure 4C*). *E. coli* cells harboring *DdTPS8*, *redB*, and *CYP508C1*, another P450 gene highly coexpressed with *DdTPS8*, produced only discoidol and no further terpenoids (*Figure 4A*). When *CYP521A1* and *redB* were expressed in the absence of *DdTPS8*, no terpene formation was observed. To test the enzyme activities in a cell-free system, a crude protein extract made from *E. coli* expressing *CYP521A1*, *redB*, and *DdTPS8* was

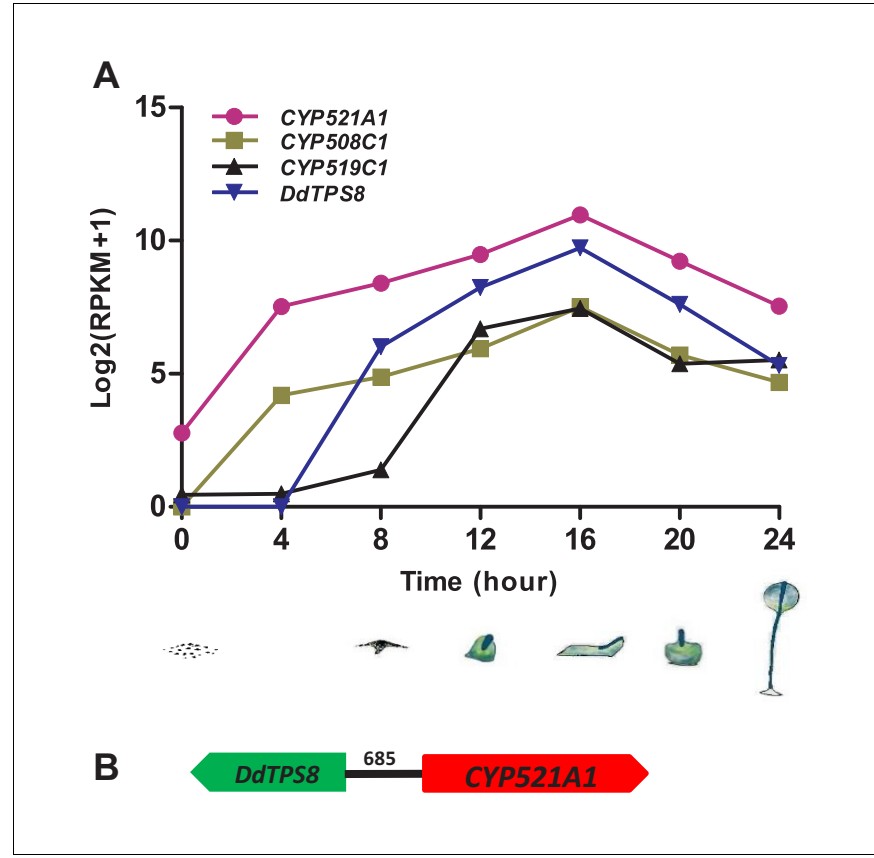

**Figure 3.** Cytochrome P450 (*CYP*) genes associated with *DdTPS8*. (**A**) Expression pattern of three *CYP* genes that showed highest level of coexpression coefficient with *DdTPS8*. The cartoons show the six stages of multicellular development of *D. discoideum*: individual cells (0 hr), streaming (8 hr), loose aggregate (10 hr), slug (16 hr), Mexican hat (20 hr) and fruiting bodies (24 hr). (**B**) *DdTPS8* and *CYP521A1* are neighbor genes. The number above the black line indicates the length of the intergenic region in base pairs.
DOI: https://doi.org/10.7554/eLife.44352.009

incubated with (*E,E*)-FDP and NADPH. Although product formation was rather low, we were able to detect discoidol and the P450 metabolite with $m/z$ = 162 in the headspace of the assay (*Figure 4—figure supplement 1*).

Production of the unknown P450 oxidation product in all of the assays tested was too low for NMR analysis. However, the molecular ion at $m/z$ = 162 in the EI mass spectrum (*Figure 5A*), pointed to a degradation of discoidol with loss of a fragment representing a molecular weight of 60 Da. The same reaction with completely labeled ($^{13}C_{15}$)discoidol, obtained from ($^{13}C_{15}$)FDP (*Rabe et al., 2015*) with discoidol synthase, produced a degradation product with incorporation of twelve $^{13}C$ atoms (*Figure 5B*), demonstrating that the degradation product is a trisnorsesquiterpene. Furthermore, the fragment ion at $m/z$ = 59 observed in the mass spectrum of discoidol (indicative of its 1-hydroxy-1-methylethyl group) was missing, suggesting that the degradation may have affected this portion of the molecule. There are two plausible hypotheses for the reaction catalyzed by CYP521A1 that would be consistent with the observed mass spectrum of the trisnorsesquiterpene (*Figure 5—figure supplement 1*). The reactive iron-oxo species of the cytochrome P450 could initiate the degradation reaction by hydrogen abstraction from the carbon originating from C-9 of FPP with formation of a stabilized allyl radical or from C-1 of FDP leading to a less stable radical (*Figure 5—figure supplement 1*). The radical intermediate then can react in the oxygen rebound by the cleavage of acetone and formation of water, resulting in different C = C double bond positions in the $C_{12}H_{18}$ product. To confirm the loss of the 1-hydroxy-1-methylethyl group and to distinguish between the alternative products, an incubation experiment with (11-$^{13}C$,1,1-$^2H_2$)FDP (*Rinkel et al.,*

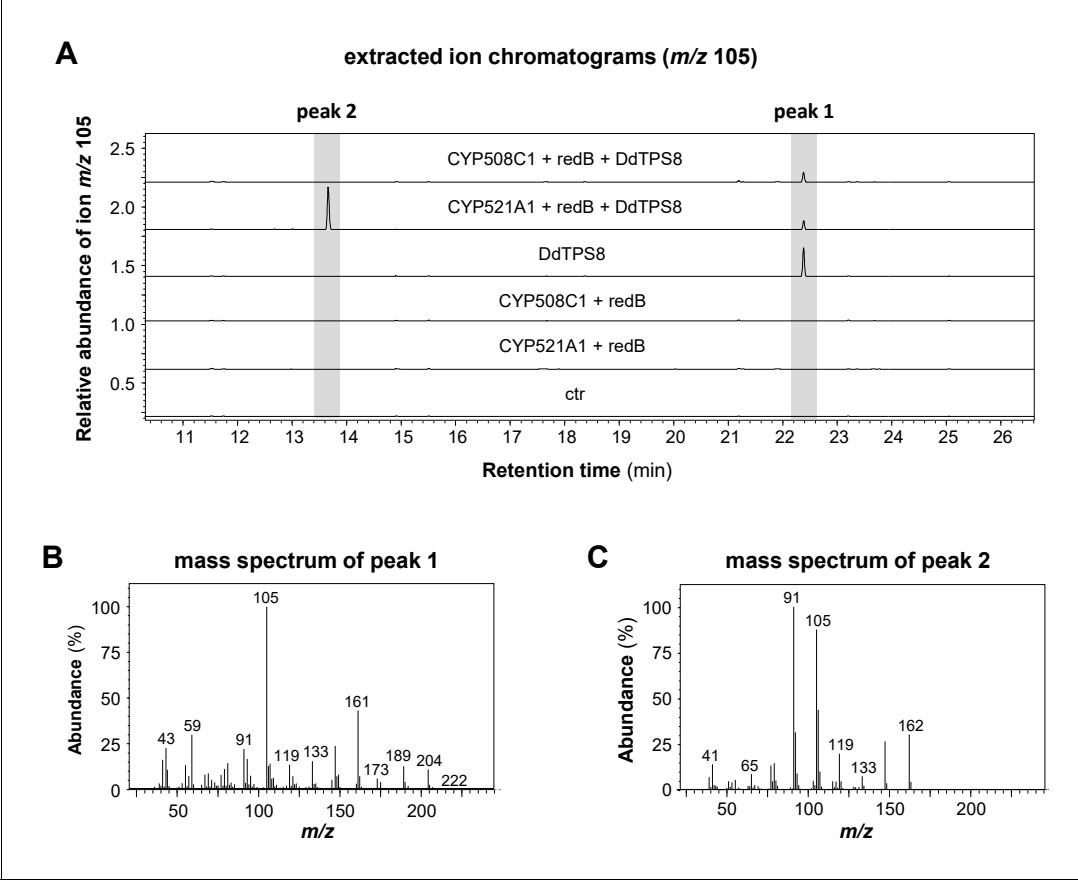

**Figure 4.** Volatile profiles of *E. coli* Bl21-DE3-Star expressing different combinations of *CYP521A1*, *CYP508C1*, the P450 reductase gene *RedB*, and the terpene synthase gene *DdTPS8*. Volatiles were collected from the headspace of the induced bacterial cultures using PDMS tubes and analyzed using GC-TDU-MS. The extracted ion chromatograms for *m/z* 105 (**A**) and the mass spectra of the DdTPS8 product (**B**) and CYP521A1 product (**C**) are shown. See also *Figure 4—figure supplement 1*.

DOI: https://doi.org/10.7554/eLife.44352.010
The following figure supplement is available for figure 4:

**Figure supplement 1.** Activity of DdTPS8 and CYP521A1 in cell-free enzyme assays in the presence of (E,E)-FDP.
DOI: https://doi.org/10.7554/eLife.44352.011

*2016*) was performed (*Figure 5—figure supplement 1*). This substrate was converted into (11-$^{13}$C,6,6-$^2$H$_2$)−**1** by the discoidol synthase and degraded by CYP521A1 into a product with a mass spectrum showing the loss of the $^{13}$C label from the 1-hydroxy-1-methylethyl group (*Figure 5C*), but retaining both deuterium atoms (*Figure 5D*). This result is consistent with the mechanism that leads to a product with a conjugated double bond system, but not with the mechanism that would result in a product with the non-conjugated double bonds (*Figure 5—figure supplement 1*). The tentatively identified compound is a new natural product for which we propose the name discodiene. A full structure elucidation of discodiene by NMR spectroscopy was not possible, because of the poor conversion of discoidol by CYP521A1. Future structure elucidation may be possible by synthesis of a reference compound.

## The *DdTPS8* knockout mutant of *D. discoideum* displayed delayed multicellular development

To examine the biological function of discodiene, cells of the *DdTPS8* mutant strain and the wild type parental AX4 strain were grown separately in HL5 medium. They were then starved, deposited on black nitrocellulose filters, and allowed to develop and their morphologies were compared. Comparing their morphologies, the mutant and the wild type developed well on black filters for the first

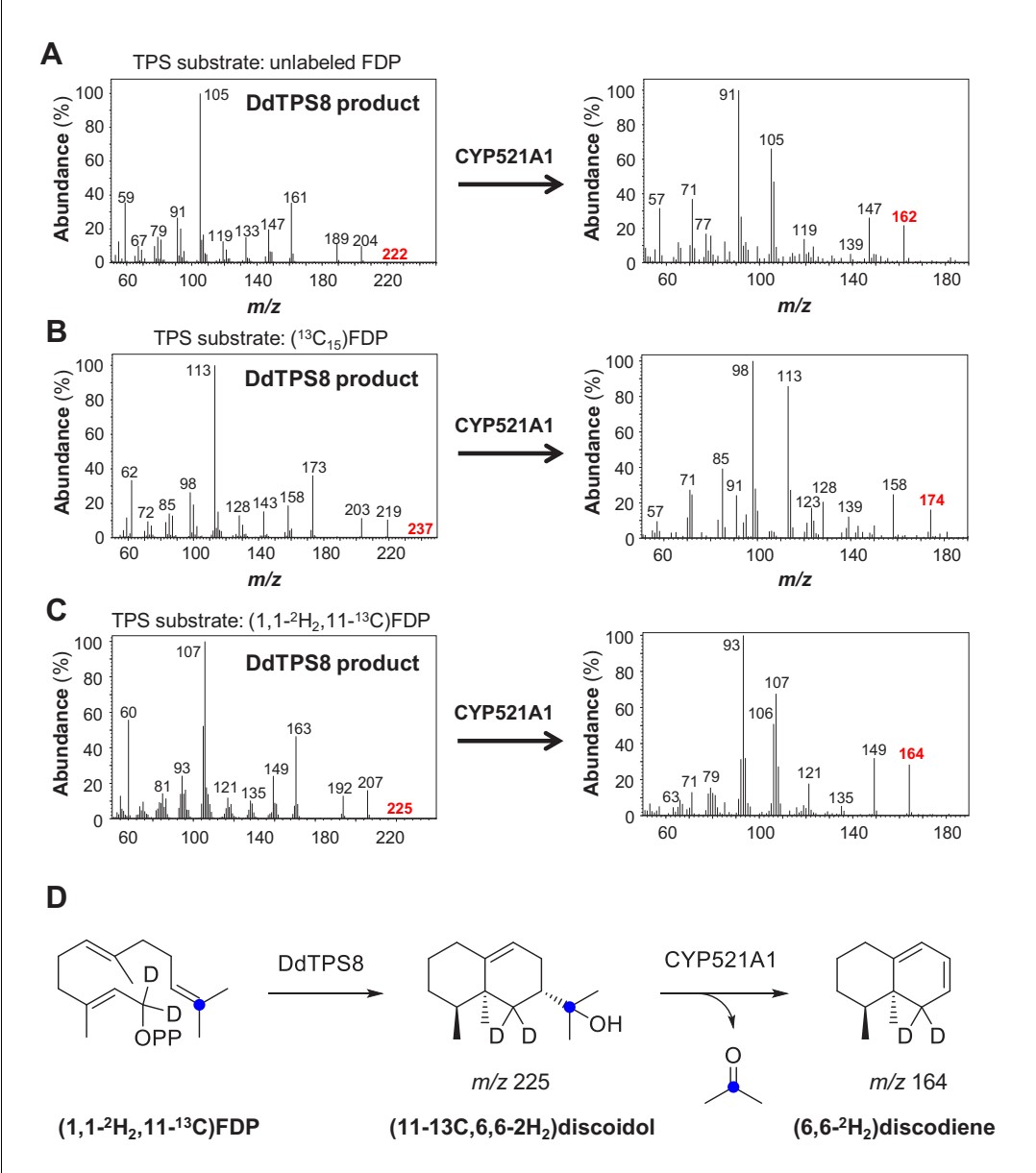

**Figure 5.** Mass spectra (**A–C**) and structures (**D**) of DdTPS8 products and CYP521A1 products derived from unlabeled farnesyl diphosphate (FDP) (**A**), 13C15-FDP (**B**), and 1,1–2 H2,11–13C-FDP (**C** + **D**). The genes were coexpressed in *E. coli* Bl21-DE3-Star together with the P450 reductase gene *RedB*. Crude protein extracts were incubated with unlabeled or labeled (*E,E*)-FDP and volatile enzyme products were collected from the headspace of the assays using PDMS tubes. Product analysis was performed with GC-TDU-MS. See also *Figure 5—figure supplement 1*.

DOI: https://doi.org/10.7554/eLife.44352.012

The following figure supplement is available for figure 5:

**Figure supplement 1.** Formation of discodiene by CYP521A1.

DOI: https://doi.org/10.7554/eLife.44352.013

few hours and there were no differences between them. However, a difference was observed after 16 hr of starvation: the multicellular development of the mutant was delayed compared to the wild type (*Figure 6*). At this time point, the wild type cells formed fingers, whereas the mutant was mainly at the tipped aggregate stage. At 20 hr (*Figure 6*), the wild type began to transition from fingers into so-called 'Mexican hats', so a mix of the two stages was observed, while the mutant was delayed at the finger stage. It is also worthwhile noting that the mutant fingers were slightly

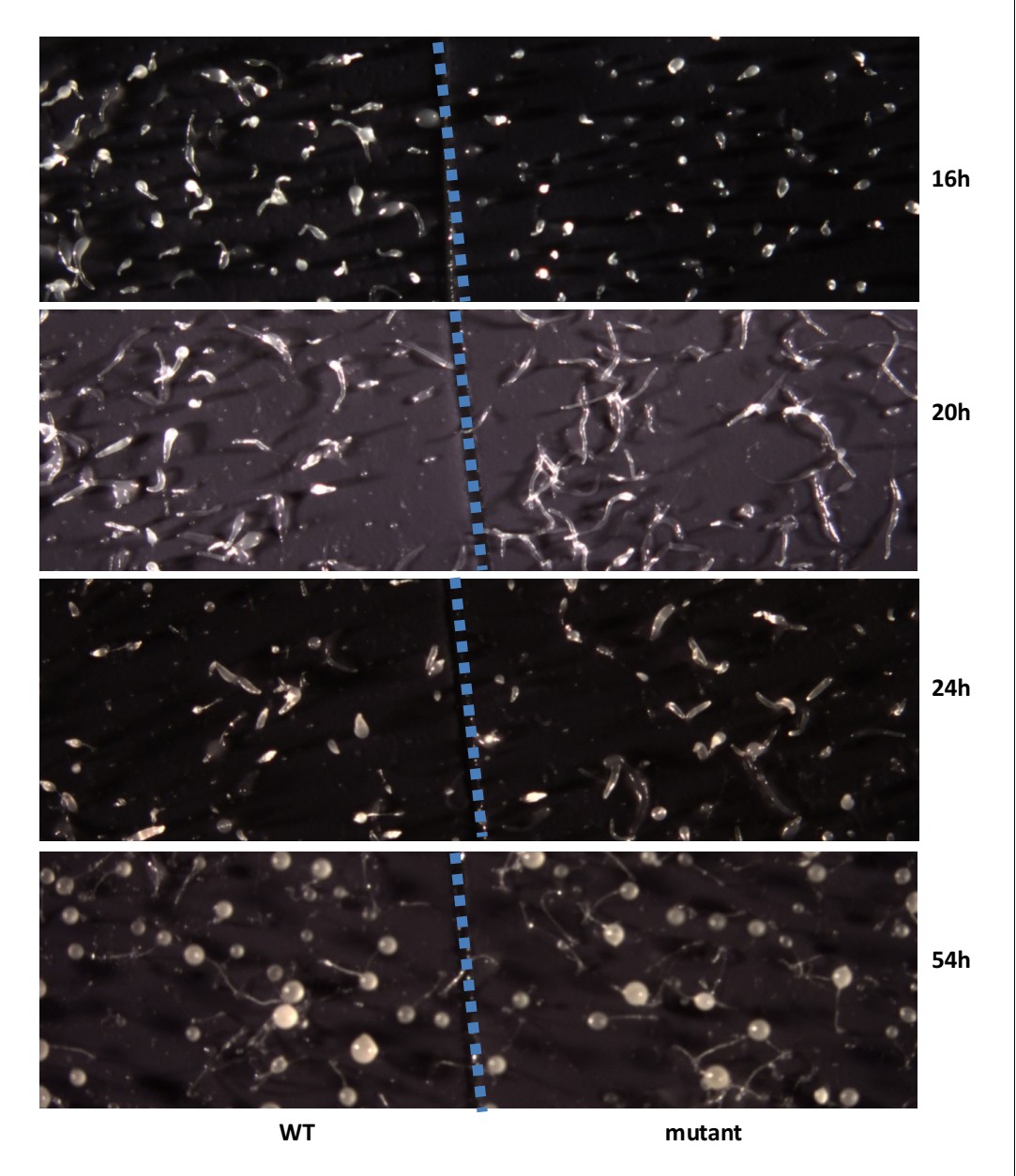

**Figure 6.** Developmental phenotype of the *DdTPS8* knockout mutant. Wild type AX4 cells (WT) and mutant *DdTPS8* cells (mutant) were grown separately in HL5 to the log phase, washed in buffer, and plated clonally on dark nitrocellulose filters. The filters were cut in half and placed next to each other in one dish. The cells were incubated in the dark at 22°C and photographed from above with a dissecting microscope at the indicated times. This experiment was independently performed three times with same results and the data shown represent one of the replicates.
DOI: https://doi.org/10.7554/eLife.44352.014

elongated and narrower compared to the wild type. At 24 hr, the wild type began entering the culmination stage, which involves complex cell movements to form a ball of spores carried on top of a cellular stalk. However, the mutant was still mostly at the finger stage with some Mexican hats visible (*Figure 6*). Eventually, both the wild type and the mutant developed into well-proportioned fruiting bodies, with stalks and spores, and they were largely indistinguishable (*Figure 6*, 54h). Spore formation and germination were also compared but no differences were observed between the wild type and the DdTPS8 mutant. These data suggest that DdTPS8 has a function during later stages of

multicellular development in *D. discoideum*, during the transition from fingers to Mexican hats, which is consistent with its temporal expression pattern (*Figure 3A*).

## Other putative *TPS-CYP* gene clusters in *D. discoideum* and *D. purpureum*

Following the functional validation of the *DdTPS8-CYP521A1* gene cluster, we examined other members of the *TPS* and *CYP* families in *D. discoideum* and identified two additional putative *TPS-CYP* gene clusters (*Figure 7A*). *DdTPS2* is organized in a head-to-head fashion with *CYP519C1* and the two are separated by an intergenic region of 580 bp. Interestingly, the opposite side of *CYP519C1* is adjacent to *DpTPS11*, which encodes a predicted partial terpene synthase (*Chen et al., 2016*). *DdTPS3* also appear to be part of a gene cluster that contains two *CYP* genes. The immediate neighbor of *DdTPS3* is a partial *CYP* gene *CYP515A2*_ps. Nevertheless, in a distance of 3148 bp from the start codon of *DdTPS3* there is an intact *CYP512A1* gene, which is arranged in a tail-to-head fashion with *DdTPS3*.

As mentioned earlier, *D. purpureum* is related to *D. discoideum* and its *TPS* family has been comprehensively characterized in our recent study (*Chen et al., 2018*). For comparison, we also analyzed the *CYP* family in *D. purpureum* and searched for putative *TPS-CYP* clusters in this species. We found a total of 54 putative *CYP* genes (*Supplementary file 4*). Among them, 47 were annotated as full-length intact genes whereas the remaining are partial genes. The 47 full-length genes were assigned to 12 families and 29 subfamilies. By examining the physical locations of *TPS* and *CYP* genes, two putative *TPS-CYP* clusters were identified in *D. purpureum*: *DpTPS5* is linked to *CYP5121A1_Dp* with an intergenic region of 612 bp and *DdTPS12* is linked to *CYP919E1_Dp* with an intergenic region of 900 bp (*Figure 7B*). In the former cluster, *DdTPS5* is in a tandem repeat with *DdTPS4*.

## Discussion

In this study, we identified and characterized a terpene synthase (*DdTPS8*)-cytochrome P450 (*CYP521A1*) gene cluster in the social amoeba *D. discoideum* (*Figure 3*) that encodes two enzymes catalyzing consecutive reactions to form the novel trisnorsesquiterpene discodiene (*Figure 4*, *Figure 5*). We also present evidence that the biosynthesis of discodiene has an effect on multicellular development of *D. discoideum* (*Figure 6*). The successful determination of CYP521A1 as discodiene synthase relied on a number of lines of evidence. First, it was clear that the sesquiterpene product of DdTPS8, discoidol (*Figure 1*), does not accumulate (*Figure 2*) and thus must be further modified

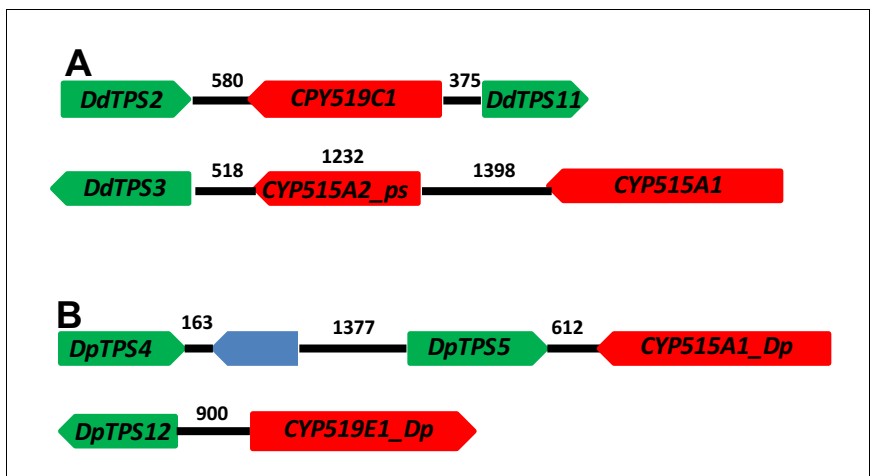

**Figure 7.** *TPS-CYP* gene clusters in *D. discoideum* (A) and *D. purpureum* (B). Green blocks depict *TPS* genes, red blocks indicate *CYP* genes. The blue block indicates a non-*TPS/CYP* gene. The numbers above the black lines indicate length in base pairs.
DOI: https://doi.org/10.7554/eLife.44352.015

after formation. Second, chemical profiling of the *DdTPS8* knockout mutant of *D. discoideum* in comparison to the wild type suggested that discoidol is converted into a degradation product (*Figure 2*). Third, coexpression analysis and colocalization in the genome implicated *CYP521A1* as the top candidate that acts downstream of *DdTPS8* (*Figure 3*). Fourth, enzyme assays confirmed that cytochrome CYP521A1 converts discoidol into a trisnorsesquiterpene that was tentatively identified by an isotopic labeling strategy as discodiene (*Figure 5*), the same compound that is produced by the wild type *D. discoideum* but not the DdTPS8 mutant (*Figure 2*).

DdTPS8 is a microbial terpene synthase that shares many mechanistic features of other typical plant terpene synthases, such as a protonation-induced cyclization of a neutral intermediate, a 1,2-hydride shift and a methyl group migration (*Degenhardt et al., 2009*). DdTPS8 catalyzes the formation of a new sesquiterpene alcohol, discoidol (*Figure 1*). It is interesting to note that the most closely related terpene synthase in *D. discoideum*, DdTPS4, also forms a sesquiterpene alcohol (*Chen et al., 2016*). However, the product of DdTPS4, (*E*)-nerolidol, is an acyclic compound in contrast to the cyclic product of DdTPS8. Based on evolutionary relatedness, the encoding genes can be inferred to have arisen from a gene duplication event. It will be interesting to determine the mechanism underlying the functional divergence of DdTPS4 and DdTPS8 in forming acyclic and cyclic sesquiterpene alcohols, respectively. It is also interesting to note that a stereoisomer of discoidol known as jinkoh-eremol has been previously isolated from an agarwood (*Aquilaria* sp.), a eudicotyledon flowering plant (*Nakanishi et al., 1983*). Since the biosynthesis of jinkoh-eremol in agarwood is most likely catalyzed by typical plant terpene synthases, which are only distantly related to the microbial type terpene synthases that include *D. discoideum* TPSs (*Jia et al., 2016*), convergent evolution has occurred in *D. discoideum* and plants for the biosynthesis of similar terpenes. Such convergence has been previously reported, as in the recent discovery that the fungal diterpene phomopsene is the product of a bacterial terpene synthase (*Lauterbach et al., 2018*). However, bacterial terpene synthases are frequently observed to make the opposite enantiomers of terpenes found in plants (*Rabe et al., 2016a*), which clearly points to different evolutionary origins in these cases.

Discodiene, the degradation product of discoidol, is a novel trisnorsesquiterpene natural product. Another important member of this group is geosmin, a particularly widespread compound in soil microorganisms that exhibits the smell of earth. While it was initially speculated that geosmin may be formed by oxidation of a terpene synthase product (*Spiteller et al., 2002*; *Cane and Watt, 2003*), it became later evident from feeding experiments (*Dickschat et al., 2005*) and characterization of the enzyme (*Jiang et al., 2007*) that a bifunctional terpene synthase catalyzes the formation of this compound involving initial cyclization to an enantiomer of hedycaryol followed by a second cyclization and oxidative cleavage of a 1-hydroxy-1-methylethyl unit. Trisnorsesquiterpenes with loss of the same 1-hydroxy-1-methylethyl unit as in discodiene have been reported from the myxobacterium *Chondromyces crocatus* (*Schulz et al., 2004*) and from streptomycetes (*Citron et al., 2012*), but the enzymology and mechanism of their formation are unclear. Oxidative dealkylations such as the reaction catalyzed by CYP521A1 converting discoidol into discodiene are known from a few other cytochrome P450s, including angelicin synthase (CYP71AJ4) from *Ammi majus* (*Larbat et al., 2009*), secologanin synthase (CYP72A1) from *Catharanthus roseus* (*Irmler et al., 2000*), DMNT/TMTT synthase (CYP82G1) from *Arabidopsis* (*Lee et al., 2010*), and the degradation of (+)-marmesin to psoralene by the psoralene synthase (*Stanjek et al., 1999*). All these enzymes are derived from plants, belonging to different CYP families. The fact that *D. discoideum* and plants belong to different kingdoms suggests convergent evolution of CYPs for catalyzing oxidative cleavage reactions.

*D. discoideum* morphogenesis is a tightly regulated developmental process that begins with the aggregation of individual cells into a mound of approximately 50,000 cells and ends with the formation of a fruiting body that consists of spores and a stalk (*Kessin, 2001*). The culmination stage, which is marked by the onset of stalk formation, occurs at the end of development, between 16 and 24 hr after the process begins. The *DdTPS8* knockout mutant exhibited delayed development at the culmination stage (*Figure 6*), around the same time that *DdTPS8* and *CYP521A1* exhibited their maximal level of expression, suggesting a potential role of discodiene in regulating *D. discoideum* development. A number of metabolites have been demonstrated to regulate *Dictyostelium* development, such as the polyketides DIF-1 (*Thompson and Kay, 2000*) and 4-methyl-5-pentylbenzene-1,3-diol (*Anjard et al., 2011*), gamma-aminobutyric acid, the nucleotide cyclic diguanylate, the cytokinin discadenine and an unidentified steroid (*Loomis, 2014*; *Schaap, 2016*). One major difference

between discodiene and these metabolites is that discodiene is a volatile. Since disrupting *DdTPS8* did not fully block the development of *D. discoideum*, discodiene might be merely a component of the volatile bouquet needed for development. It is also possible that discodiene may not control developmental programs directly, but rather influence them indirectly by acting as an 'aggregation/culmination pheromone' as already speculated for volatile terpenoids in *D. discoideum* (*Chen et al., 2016*).

The organization of terpenes synthase genes and *CYP* genes involved in the same biochemical pathway as metabolic gene clusters has been documented in a growing number of cases including bacteria (*Nett et al., 2017*), fungi (*Quin et al., 2014*), and plants (*Boutanaev et al., 2015*). Our findings add the *Dictyostelium* social amoebae to this list. With only 9 *TPS* genes in *D. discoideum*, it is unlikely that the clustering of with *CYP* genes (*Figure 3B* and *Figure 7A*) is accidental. It is equally intriguing to observe that *D. purpureum* contains two *TPS-CYP* clusters (*Figure 7B*). With the evolutionary relationship of *TPS* genes from *D. discoideum* and *D. purpureum* recently determined (*Chen et al., 2018*), we are able to examine the relatedness of these *TPS-CYP* gene clusters. The ortholog of *DdTPS8* in *D. purpureum* is *DpTPS6* (*Chen et al., 2018*), which is not clustered with any *CYP* gene. Moreover, *D. purpureum* lacks an ortholog of *CYP521A1*, implying that the *D. purpureum* may not produce discodiene, consist with the volatile profiling of this species (*Chen et al., 2018*). *DdTPS2-CYP519C1* and *DpTPS2-CYP919E1_Dp* appear to be species-specific. In contrast, *DdTPS3-CYP515A1* and *DpTPS5-CYP515A1_Dp* are an orthologous pair based on the orthology of the respective *TPS* genes (*DdTPS3* and *DpTPS5*) (*Chen et al., 2018*) and *CYP* genes (*CYP515A1* and *CYP515A1-Dp* are in the same subfamily). Taken together, these *TPS-CYP* clusters suggest the biosynthesis of both shared as well as species-specific terpenoids in *D. discoideum* and *D. purpureum*.

Besides terpenoids, dictyostelid social amoebae produces many other types of natural products (*Barnett and Stallforth, 2018*), including polyketides. Polyketide synthases (PKSs) are pivotal enzymes for polyketide biosynthesis (*Khosla, 2009*) and the production of some polyketides involves CYPs (e.g., *Bedewitz et al., 2018*). The *D. discoideum* genome contains 40 *PKS* genes (*Zucko et al., 2007*). With the demonstration of the *DdTPS8-CYP521A1* cluster catalyzing consecutive reactions to produce a novel natural product, the potential for generating chemical diversity of natural products in *D. discoideum* using the building blocks derived from 9 TPSs, 41 CYPs and 40 PKSs is enormous, which can be envisioned to be a fruitful future research direction. It would be interesting to know what functions these natural products have, especially in chemical communications between the organism and its environment.

# Materials and methods

## Strains of *D. discoideum* and mutant analysis

The wild type AX4 strain of *D. discoideum* (DBS0237637) was obtained from the Dictybase Stock Center (http://dictybase.org/). The *DdTPS8* mutant was obtained from the gridded collection of the Genome Wide *Dictyostelium* Insertion (GWDI) Project (https://remi-seq.org). The mutant stain was cultured clonally and the genotype was verified by diagnostic PCR with insertion (Blasticidin resistance gene) specific and gene specific primers (*Supplementary file 5*), followed by characterization with gene-insert specific primers to verify the location of the insertion. To characterize the developmental progression of the mutant, we grew both wild type (AX4) and *DdTPS8* mutant in HL5 liquid medium to the mid-log growth phase. We then collected the cells by centrifugation, washed them and deposited them on black nitrocellulose filters (*Shaulsky and Loomis, 1993*). We examined developmental morphology and photographed the structures from above using a dissecting microscope. This experiment was repeated three times.

## Headspace collection and GC-MS analysis

Culturing of the wild type AX4 and the *DdTPS8* mutant strain of *D. discoideum*, headspace collection and chemical identification using GC-MS were performed as previously reported (*Chen et al., 2016*) with three biological replicates.

## CYP gene search and gene co-expression analysis

Proteome sequences of *D. discoideum* and *D. purpureum* were downloaded from Dictybase (http://dictybase.org) and were used as dataset for identifying *CYP* genes. A HMM model of P450 gene family (PF00067) downloaded from Pfam 31.0 (http://pfam.xfam.org) was used to search putative p450 genes against the downloaded protein dataset using HMMER 3.1b two with an e-value cutoff of $1e^{-2}$. CYPs were blast searched against named CYPs from all protists. Sequences were sorted based on best blast hit percent ID and named based on the Standardized Cytochrome P450 nomenclature rules (*Nelson et al., 1996*).

Expression data of *CYP* genes in *D. discoideum* during its 24 hr developmental program were obtained from online web-interface program DictyExpress (https://dictyexpress.research.bcm.edu). Pearson correlation coefficients (PCCs) between the expression of *DdTPS8* and that of *CYP* genes of *D. discoideum* were calculated using IBM SPSS (v.25, https://www.ibm.com).

## Cloning of two *CYP* genes, *redB* gene and vector construction

Full-length cDNAs for two *CYP* genes, *CYP521A1* and *CYP508C1*, and one cytochrome p450 reductase gene (DDB_G0269912, known as *redB*) were cloning by RT-PCR. Approximately 0.1 g tissue of *D. discoideum* at the culmination stage was collected and disrupted by TissueLyser II (https://www.qiagen.com). Total RNA was isolated using RNeasy Plant Mini Kit (https://www.qiagen.com) and converted to cDNAs using the First strand cDNA synthesis kit (https://www.gelifesciences.com). Full-length cDNAs for each of the three genes were amplified by PCR using gene-specific primers (*Supplementary file 5*), cloned into pGEM-T Easy vector (https://www.promega.com), and fully sequenced. Next, the redB gene was added with the *NdeI* and *KpnI* restriction sites using PCR and ligated into the *NdeI* and *KpnI* site of the second MSC of the vector pRSFDuet1 (http://www.emd-millipore.com) to produce pRSFDuet1::*redB*. Then *CYP521A1* and *CYP508C1* were cloned into the *BamH1* and *PstI* sites of the first MSC of pRSFDuet1::*redB* to produce two plasmids pRSFDuet1:: CYP521A1::*redB* and pRSFDuet1:: CYP508C1::*redB*.

## DdTPS8 protein purification and product isolation

A full-length cDNA for *DdTPS8* was cloned into the pET32a vector. *E. coli* BL 21 harboring pET32a_*DdTPS8* was inoculated in a LB liquid preculture containing ampicillin (50 mg/L). For protein isolation the preculture was used to inoculate large scale 2YT liquid cultures ($8 \times 1$ L) containing ampicillin (50 mg/L). Cells were grown to an $OD_{600} = 0.4$ at 37°C and 160 rpm, followed by cooling of the cultures to 18°C. IPTG (0.4 mM) was added and the culture was further incubated at 18°C and 160 rpm over night. Cells were harvested by centrifugation at 4°C and 3600 rpm for 30 min. The supernatant was discarded and the cell pellet was resuspended in 80 mL binding buffer (20 mM $Na_2HPO_4$, 0.5 M NaCl, 20 mM imidazole, 1 mM $MgCl_2$, pH 7.0). Cell disruption was done by ultra-sonication on ice for $8 \times 60$ s. The cell debris was removed by repeated centrifugation ($2 \times 10$ min) at 4°C and 11000 rpm to yield the soluble enzyme fractions. Protein purification was performed by $Ni^{2+}$-NTA affinity chromatography with $Ni^{2+}$-NTA superflow (Qiagen) using binding buffer ($4 \times 20$ mL; 20 mM $Na_2HPO_4$, 0.5 M NaCl, 20 mM imidazole, 1 mM $MgCl_2$, pH 7.0) and elution buffer ($4 \times 20$ mL; 20 mM $Na_2HPO_4$, 0.5 M NaCl, 20 mM imidazole, 1 mM $MgCl_2$, pH 7.0). The enzyme fractions were used for enzyme reactions with the natural substrate FDP (50 mg, final concentration c = 0.2 mg/mL). The incubation experiment was performed at 28°C for 3 hr and was extracted three times with hexane. The combined organic layers were dried over $MgSO_4$ and concentrated under reduced pressure. Column chromatography on silica gel (pentane: diethyl ether 5:1) yielded the terpene alcohol (5.0 mg) as colorless oil.

## *E. coli* expression and headspace analysis

The two plasmids pEXP5-CT/TOPO::*DdTPS8* and pRSFDuet1:: *CYP:: redB* were cotransformed into *E. coli* Bl21. The resultant strain was cultured overnight and then induced for protein expression by adding isopropyl β-D-1-thiogalactopyranoside. Volatiles emitted from the *E. coli* culture were collected by solid phase microextraction for 1 hr, and analyzed by GC-MS. Protein expression and headspace collections were repeated three times.

## Enzyme reactions with isotopically labeled substrates

Incubation experiments with DdTPS8 and labeled substrates were performed with the pure protein fractions obtained from 300 mL *E. coli* BL 21 pET32a_*DdTPS8* liquid culture as reported above. All incubation experiments were done with the isotopically labeled substrate (0.6–1 mg, c = 0.1 mg/mL) at 28°C for 3 hr. The reaction mixtures were extracted with benzene-d6 (0.8 mL), the organic layers were separated and directly analyzed by GC/MS and NMR. For the determination of the absolute configuration of the DdTPS8 product (R)-(1-$^{13}$C,1-$^{2}$H) and (S)-(1-$^{13}$C,1-$^{2}$H)geranyl diphosphate (each 0.8 mg) were elongated with FDP synthase and isopentenyl diphosphate (0.5 mg) in the same reaction vessel and extracted after 3 hr incubation at 28°C.

## Acknowledgements

We thank Christopher Thompson and the GWDI team for providing the *DdTPS8* mutant before the public release of the mutant collection and Dr. Guo Wei for discussions about CYP enzyme assays.

## Additional information

### Funding

| Funder | Author |
| --- | --- |
| University of Tennessee Institute of Agriculture | Feng Chen |

The funders had no role in study design, data collection and interpretation, or the decision to submit the work for publication.

### Author contributions

Xinlu Chen, Conceptualization, Resources, Formal analysis, Validation, Investigation, Methodology, Writing—original draft, Writing—review and editing; Katrin Luck, Patrick Rabe, Resources, Data curation, Formal analysis, Validation, Investigation, Methodology, Writing—original draft, Writing—review and editing; Christopher QD Dinh, Resources, Data curation, Validation, Investigation, Visualization, Writing—original draft, Writing—review and editing; Gad Shaulsky, Conceptualization, Resources, Data curation, Supervision, Investigation, Methodology, Writing—original draft, Project administration, Writing—review and editing; David R Nelson, Data curation, Formal analysis, Validation, Investigation, Methodology, Writing—original draft, Writing—review and editing; Jonathan Gershenzon, Conceptualization, Data curation, Supervision, Funding acquisition, Writing—original draft, Project administration, Writing—review and editing; Jeroen S Dickschat, Conceptualization, Resources, Data curation, Formal analysis, Supervision, Validation, Investigation, Methodology, Writing—original draft, Project administration, Writing—review and editing; Tobias G Köllner, Conceptualization, Resources, Data curation, Supervision, Validation, Investigation, Visualization, Writing—original draft, Project administration, Writing—review and editing; Feng Chen, Conceptualization, Data curation, Supervision, Funding acquisition, Investigation, Writing—original draft, Project administration, Writing—review and editing

### Author ORCIDs

Xinlu Chen http://orcid.org/0000-0002-7560-6125
Gad Shaulsky https://orcid.org/0000-0002-0532-0551
David R Nelson http://orcid.org/0000-0003-0583-5421
Jeroen S Dickschat http://orcid.org/0000-0002-0102-0631
Feng Chen http://orcid.org/0000-0002-3267-4646

### Decision letter and Author response

Decision letter https://doi.org/10.7554/eLife.44352.025
Author response https://doi.org/10.7554/eLife.44352.026

## Additional files

### Supplementary files

• Supplementary file 1. NMR data of discoidol recorded in $C_6D_6$.
DOI: https://doi.org/10.7554/eLife.44352.016

• Supplementary file 2. Cytochrome p450 genes in *Dictyostelium discoideum*.
DOI: https://doi.org/10.7554/eLife.44352.017

• Supplementary file 3. *CYP* genes of *Dictyostelium discoideum* that show significant coexpression with *DdTPS8*.
DOI: https://doi.org/10.7554/eLife.44352.018

• Supplementary file 4. Cytochrome p450 genes in *Dictyostelium purpureum*.
DOI: https://doi.org/10.7554/eLife.44352.019

• Supplementary file 5. Primers used in gene cloning and vector construction.
DOI: https://doi.org/10.7554/eLife.44352.020

• Transparent reporting form
DOI: https://doi.org/10.7554/eLife.44352.021

### Data availability

The sequence for CYP521A1 has been deposited in the GenBank database (accession no. MH923436).

The following dataset was generated:

| Author(s) | Year | Dataset title | Dataset URL | Database and Identifier |
|---|---|---|---|---|
| Chen X, Luck K, Rabe P, Quang Dung Dinh C, Shaulsky G, Nelson DR, Gershenzon J, Dickschat JS, Köllner TG, Chen F | 2019 | Sequence for CYP521A1 | https://www.ncbi.nlm.nih.gov/nuccore/mh923436 | NCBI Nucleotide, MH923436 |

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
