## [Decision Letter]

Thank you for submitting your article "A Terpene Synthase-Cytochrome P450 Cluster in *Dictyostelium* Produces a Novel Norsesquiterpene and Regulates Development" for consideration by *eLife*. Your article has been reviewed by Detlef Weigel as the Senior Editor, a Reviewing Editor, and two reviewers. One of the reviewers, René Feyereisen, has agreed to reveal his identity.

The reviewers have discussed the reviews with one another and the Reviewing Editor has drafted this decision to help you prepare a revised submission.

The manuscript by Chen and colleagues makes key advances in understanding the role of low molecular weight signals in the biology of Dictyostelium. The study demonstrates that this amoeba produces terpenes, at least one of which is converted into a novel terpenoid via a cytochrome P450 and suggests that it may influence the timing of multicellular development. These results provide the first biochemical characterization of a terpene synthase – P450 cluster in amoebas.

We agreed that the data are important and deserve publication. However, we are not convinced of the biological relevance of the developmental effect attributed to discodiene, given the small effect size of the DdTPR8 mutant. Moreover, you obtained your mutant via random insertion and you did not attempt to rule out that the phenotype was only pleiotropic (e.g. via complementing the mutant with the wild-type gene and test whether this restores normal development by generating a second knock-out by homologous recombination). This part of your manuscript should therefore be significantly reduced and toned down to be acceptable to *eLife*. Finally, we also think that there might be alternative scenarios for the role and formation of discodiene worthwhile discussing. That is why we ask you for the following:

1) The effect of discodiene on the development of the amoeba must be substantiated by demonstrating that it is not only pleiotropic. If not, it can only be cursorily mentioned. In that case "and regulates multicellular development" should be removed from the Title.

2) The effects of loss of DdTPS8 gene function on development are rather minor (so assuming they are not pleiotropic). A few hours difference in developmental progression also easily occurs on a day to day base for wild-type Dictyostelium. Therefore, you should clearly acknowledge that the biological significance of this individual terpene may be rather small. However, we suggest you to also mention that discodiene might be merely a component of a (terpene) bouquet needed for development and that knocking out just one may not be enough to fully block whatever role these volatiles may play in Dictyostelium development (or other aspects of their life history). Finally, you should acknowledge that discodiene may not control developmental programs directly (if at all) but may influence these indirectly by acting as facilitator, or "aggregation pheromone".

3) While we judge that the role of discodiene in development is overstated (see 1 and 2), we also feel that the novelty of a functional biosynthetic cluster of terpene synthase – P450 in this organism (including the uncommon reaction catalyzed by the P450) is understated. Dictyostelium has 40 PKS, 41 P450 and 9 TPS genes: the potential for generating chemical diversity in "synthetic biology" using these building blocks is therefore enormous, and because this is not a plant, a fungus or a bacterium, diversity can be assumed (and indeed, discodiene is itself a previously unreported (nor)sesquiterpene). We believe that there is sufficient reason to emphasize the novelty from the perspective of the fields of natural products – chemical ecology much more while being much more cautious about its role in developmental programs.

4) There was much discussion about the robustness of the data in Figure 6, because it was not immediately clear that these experiments had actually been repeated three times. This is too much hidden in the Materials and methods section. Mention and discuss the reproducibility of these results explicitly (and also mention this in the figure legend).

5) In the first paragraph it is stated that apart from cAMP no low molecular weight signaling molecules have been reported to be active in Dictyostelium development. This is incorrect. There are at least two polyketides, DIF-1 (Thompson and Kay, 2000) and MBPD, with essential developmental roles. Other low molecular weight compounds required for stalk and spore differentiation are GABA, the nucleotide c-di-GMP, the cytokinin discadenine and un unidentified steroid (see the Loomis, 2014 and Schaap, 2016 reviews). This must be mentioned to place the discodiene story in the right context.

6) We propose an alternative mechanism for the carbon-carbon scission yielding acetone and discoidiene i.e. P450 abstraction of the C8 α hydrogen, leading to a discoidol C8 radical that would rearrange by scission of the adjacent isopropanol group to a radical. This isopropanol radical would react in the classical oxygen rebound with the [Fe-OH]3+, and loss of water to acetone. This may be more likely that the abstraction of the hydroxyl hydrogen and regioselective rearrangement of the radical following carbon-carbon cleavage. The alternative mechanism would be consistent with the retention of deuterium and would replace a relatively difficult selective rebound on the hypothetical C7 radical (with removal of a C8 but not C6 hydrogen), by a probably favored rebound on the isopropanol radical. Such an alternative mechanism has been proposed for the analogous reactions of CYP71AJ1 and CYP71AJ4, with modeling of CYP71AJ1 with marmesin (Larbat et al., 2007) specifically showing the optimal positioning of the carbon α to the isopropyl group (i.e. C8 of discoidol). Similarly, in the CYP82G1 model, the C5 carbon of the nerolidol and geranyllinalool substrates (equivalent to C8 of discoidol) is ideally positioned for hydrogen abstraction, whereas the C3 hydroxyl group is predicted to hydrogen bond with an active site threonine. If an analogous substrate positioning and hydrogen bonding could be shown in CYP521A1, then this would favor oxygen rebound on the isopropanol radical.

---

## [Author Response]

1) The effect of discodiene on the development of the amoeba must be substantiated by demonstrating that it is not only pleiotropic. If not, it can only be cursorily mentioned. In that case "and regulates multicellular development" should be removed from the Title.

While the DdTPS8 knockout mutant displayed delayed development, we agree with the reviewers that it is uncertain at this time whether discodiene has roles in other processes in development or functions in chemical communications of the amoeba with other organisms. With this acknowledgement, we gladly accepted the suggestion to tone down the role of discodiene in controlling/regulating development, which is reflected in the revised Abstract, Results section and Discussion section. “And regulates multicellular development” has been removed from the title as suggested.

2) The effects of loss of DdTPS8 gene function on development are rather minor (so assuming they are not pleiotropic). A few hours difference in developmental progression also easily occurs on a day to day base for wild-type Dictyostelium. Therefore, you should clearly acknowledge that the biological significance of this individual terpene may be rather small. However, we suggest you to also mention that discodiene might be merely a component of a (terpene) bouquet needed for development and that knocking out just one may not be enough to fully block whatever role these volatiles may play in Dictyostelium development (or other aspects of their life history). Finally, you should acknowledge that discodiene may not control developmental programs directly (if at all) but may influence these indirectly by acting as facilitator, or "aggregation pheromone".

We agree with the comments. The following sentences have been added to the revised Discussion section:

“Since disrupting DdTPS8 did not fully block the development of D. discoideum, discodiene might be merely a component of the volatile bouquet needed for development. It is also possible that discodiene may not control developmental programs directly, but rather influence them indirectly by acting as an "aggregation/cultimation pheromone" as already speculated for volatile terpenoids in D. discoideum (Chen et al., 2016).”

3) While we judge that the role of discodiene in development is overstated (see 1 and 2), we also feel that the novelty of a functional biosynthetic cluster of terpene synthase – P450 in this organism (including the uncommon reaction catalyzed by the P450) is understated. Dictyostelium has 40 PKS, 41 P450 and 9 TPS genes: the potential for generating chemical diversity in "synthetic biology" using these building blocks is therefore enormous, and because this is not a plant, a fungus or a bacterium, diversity can be assumed (and indeed, discodiene is itself a previously unreported (nor)sesquiterpene). We believe that there is sufficient reason to emphasize the novelty from the perspective of the fields of natural products – chemical ecology much more while being much more cautious about its role in developmental programs.

This point is well taken. We added one full paragraph in the Discussion section to illustrate the potential untapped diversity of natural products in Dictyostelium that could result from the combinational assembly of its TPSs, P450s and PKSs. The revised Abstract is now ended with one added sentence to highlight this novelty.

4) There was much discussion about the robustness of the data in Figure 6, because it was not immediately clear that these experiments had actually been repeated three times. This is too much hidden in the Materials and methods section. Mention and discuss the reproducibility of these results explicitly (and also mention this in the figure legend).

The times of independent repeats for the data in Figure 6 is now specified in Materials and methods section. We also added this information to the legend for Figure 6 as suggested.

5) In the first paragraph it is stated that apart from cAMP no low molecular weight signaling molecules have been reported to be active in Dictyostelium development. This is incorrect. There are at least two polyketides, DIF-1 (Thompson and Kay, 2000) and MBPD, with essential developmental roles. Other low molecular weight compounds required for stalk and spore differentiation are GABA, the nucleotide c-di-GMP, the cytokinin discadenine and un unidentified steroid (see the Loomis, 2014 and Schaap, 2016 reviews). This must be mentioned to place the discodiene story in the right context.

We apologize for this omission. The incorrect statement in the Introduction is now removed. Instead, we added the expanded information about other low molecular weight metabolites with defined roles in development to the Discussion section when discussing the biological role of discodiene.

6) We propose an alternative mechanism for the carbon-carbon scission yielding acetone and discoidiene i.e. P450 abstraction of the C8 α hydrogen, leading to a discoidol C8 radical that would rearrange by scission of the adjacent isopropanol group to a radical. This isopropanol radical would react in the classical oxygen rebound with the [Fe-OH]3+, and loss of water to acetone. This may be more likely that the abstraction of the hydroxyl hydrogen and regioselective rearrangement of the radical following carbon-carbon cleavage. The alternative mechanism would be consistent with the retention of deuterium and would replace a relatively difficult selective rebound on the hypothetical C7 radical (with removal of a C8 but not C6 hydrogen), by a probably favored rebound on the isopropanol radical. Such an alternative mechanism has been proposed for the analogous reactions of CYP71AJ1 and CYP71AJ4, with modeling of CYP71AJ1 with marmesin (Larbat et al., 2007) specifically showing the optimal positioning of the carbon α to the isopropyl group (i.e. C8 of discoidol). Similarly, in the CYP82G1 model, the C5 carbon of the nerolidol and geranyllinalool substrates (equivalent to C8 of discoidol) is ideally positioned for hydrogen abstraction, whereas the C3 hydroxyl group is predicted to hydrogen bond with an active site threonine. If an analogous substrate positioning and hydrogen bonding could be shown in CYP521A1, then this would favor oxygen rebound on the isopropanol radical.

We are highly appreciative of the suggested alternative mechanism catalyzed by CYP521A1. Now we agree that it is a more possible reaction mechanism. This mechanism is now added to our revised manuscript as a part of Figure 5—figure supplement 1 and properly discussed.